# Modeling Directional Brightness Temperature (DBT) over Crop Canopy with Effects of Intra-Row Heterogeneity

**Yongming Du** [1], **Biao Cao** [1,*], **Hua Li** [1], **Zunjian Bian** [1], **Boxiong Qin** [1,2], **Qing Xiao** [1,2], **Qinhuo Liu** [1,2], **Yijian Zeng** [3] **and Zhongbo Su** [3]

1 State Key Laboratory of Remote Sensing Science, Aerospace Information Research Institute, Chinese Academy of Sciences, Beijing 100101, China; duym@radi.ac.cn (Y.D.); lihua@radi.ac.cn (H.L.); bianzj@radi.ac.cn (Z.B.); qinbx@radi.ac.cn (B.Q.); xiaoqing@radi.ac.cn (Q.X.); liuqh@radi.ac.cn (Q.L.)
2 University of Chinese Academy of Sciences, Beijing 100049, China
3 Faculty of Geo-Information Science and Earth Observation (ITC), University of Twente, 7500 AE Enschede, The Netherlands; y.zeng@utwente.nl (Y.Z.); z.su@utwente.nl (Z.S.)
* Correspondence: caobiao@radi.ac.cn

**Abstract:** In order to improve the simulation accuracy of directional brightness temperature (DBT) and the retrieval accuracy of component temperature, a model considering intra-row heterogeneity to simulate the DBT angular distribution over crop canopy is proposed. At individual scale, the probability of leaf appearance is inversely proportional to the distance from central stem. On the basis of this assumption, we formulated leaf area volume density (LAVD) spatial distribution at three hierarchical scales: individual scale, row scale, and scene scale. The equations for directional gap probability and bi-directional gap probability were modified to adapt the heterogeneity of row structure. Afterwards, a straightforward radiative transfer model was built based on the gap probabilities. A set of simulated data was generated by the thermal radiosity-graphics combined model (TRGM) as the benchmark to evaluate both forward simulation and inversion ability of the new model; we compared the new DBT model against an existing model assuming row as homogeneous box. With the growth of crops, the canopy structure of row crops will gradually change from row structure to continuous canopy. The new DBT model agreed with the TRGM model much better than the homogeneous row model at the middle stage of the crop growth season. The new model and the homogeneous row model achieve similar accuracy at early stage and end stage. At the middle growth stage, the new model can improve the accuracy of soil temperature retrieval. We recommend the new DBT model as an option to improve the DBT simulation and component temperature retrieval for row-planted crop canopy. In particular, the more accurate component temperatures during the middle growth stage are fundamentally important in characterizing crop water status, evapotranspiration, and soil moisture, which are subsequently critical for predicting crop productivity.

**Keywords:** directional brightness temperature (DBT); modeling; row crop canopy; intra-row heterogeneity; component temperature retrieval

## 1. Introduction

Thermal infrared (TIR) remote sensing has wide applications for crops. The surface temperature acquired efficiently by TIR greatly influences the soil and plant respiration rates, which are of great concern in carbon cycles [1–4]. TIR can give early detection of plant disease prior to being detectable by visual and near infrared (V-NIR) bands [5,6]. TIR is used to characterize crop water status by means of combination index, such as "crop water stress index" (CWSI) [7,8], and "water deficit index" (WDI) [9].

TIR data are widely used in land surface temperature (LST) and energy balance and evapotranspiration (ET) estimation at regional and global scales [10–13].

Directional anisotropy of thermal radiation has been observed from ground-based [14], air-born [15–17], and space-born [13,18,19] instruments. The deviation of observation from different angles degrades the reliability of TIR data and obstructs applications. On the other hand, TIR directional anisotropy provides a possibility to separate component temperatures (CTs) from the multi-angle data [20–24]. To well understand the underlying mechanism of the TIR directional anisotropy, directional brightness temperature (DBT) models are required to link the directional radiance and the illumination viewing geometry, vegetation architecture, and functioning. A high-precision DBT model is a prerequisite for both angular correction and CT inversion.

DBT models fall into three categories: radiative transfer (RT) model, geometric optics (GO) model, and computer model. The RT model, represented by 4SAIL model, was designed for homogeneous scene [25]. Crops are often planted in rows owing to mechanical sowing operation and irrigation management. Row structure greatly affects the DBT angular distribution at early growth stages. GO models have natural advantage in simulating the angular variation of DBT by calculating the mutual shading between components in both the view and illuminate directions. Most published DBT models for row crops can be categorized as GO models. Jackson et al. (1979) [26] and Kimes (1983) [27] abstracted the row-planted crop canopy as a solid box without gaps. Chen et al. (2002) [28] and Yan et al. (2003) [29] introduced bidirectional gap probability to characterize the porous rows. Yu et al. (2004) [1] improved the position of hot spot of DBT over row-planted maize by lifting the canopy a distance above the soil. Du et al. (2007) [30] added the effect of ear layer for wheat at the mature stage. Computer models can reproduce the physical processes of TIR radiative transfer within a near-real scene by a ray tracing method, for example, DART [31], or a radiosity approach, for example, TRGM [32]. They can provide very high simulation precision at the cost of being time-consuming.

Luquet et al. (2004) [33] worked out that the TIR directional anisotropy bring errors of 14% near nadir and more than 40% for larger view angle in WDI for sparse crops. The sunlit soil effects greatly influenced the uncertainty of WDI. Colaizzi et al., (2016) [34] calculated the sunlit and shaded soil heat flux separately under row crops. The two source energy balance model (TSEB) needs more precise CT as input [35]. These applications show that more accurate DBT models are needed.

Up to now, all the DBT models assumed that leaves were homogeneous in rows. In fact, this assumption cannot hold in the case of row crop. The density of leaves in the row reaches a maximum in the center, and decreases gradually with the distance from the center. The present paper aims to analyze the effects of the intra-row heterogeneity on both forward DBT simulation and CT inversion. The aim of this paper is to improve both the forward simulation accuracy and CT inversion accuracy by considering the intra-row heterogeneity.

The rest of the paper is organized as follows. Section 2 describes the new DBT model considering intra-row heterogeneity. Section 3 gives both an evaluation of the forward simulation and an evaluation of the CT inversion of the new model using synthetic data. Section 3 also illustrates the model's performance spanning the whole crop growth season. Section 4 discusses the sensitivity of input parameters, potential applications of the new model, and the limitation of the model. Section 5 concludes the whole paper.

## 2. Methods

### 2.1. Modeling Leaf Area Volume Density (LAVD) Distribution

Row crop field is a typical scene. The existing row models assumed the leaves were arranged uniformly in rows [1,36,37]. In fact, LAVD is high at the center of an individual plant and decreases gradually with the distance from the center in a centrosymmetric manner, for example, maize, rice, wheat, and other similar crops. On this basis, we formulate the leave area volume density (LAVD) space distribution at individual scale, row scale, and scene scale hierarchically.

For convenience of discussion at the individual scale, a cylindrical coordinate system is introduced (Figure 1b). The origin of the cylindrical coordinate system is set at the intersection of stem and soil plane. The z-axis overlaps the central line of stem. The azimuth starts from the row direction. The radius is denoted as $\rho$ and the azimuth is denoted $\varphi$. The z-axis is denoted as $z$.

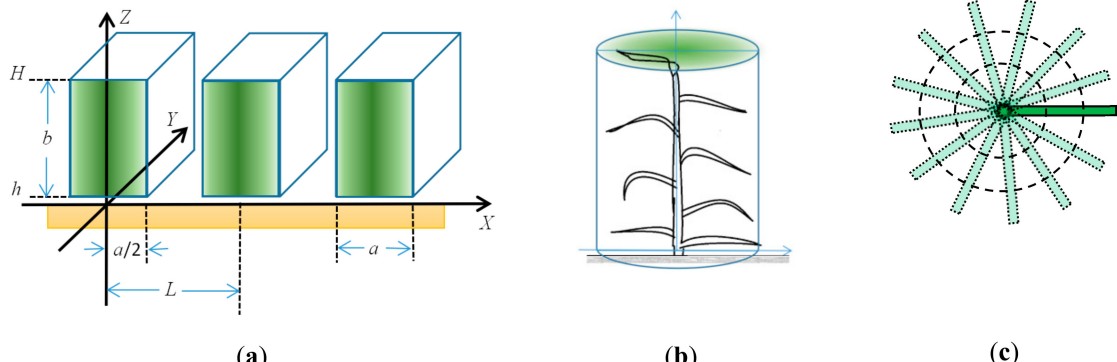

|     |     |     |
| --- | --- | --- |
| (**a**) | (**b**) | (**c**) |

**Figure 1.** Coordinate system for row crop scene and for an individual plant. (**a**) The Cartesian coordinate system for the row structure. *L* is the distance between two neighboring rows, *a* is the width of row, *b* is the thickness of row, and *H* and *h* are the heights of upper and bottom edge of row. (**b**) The cylindrical coordinate system for individual scale. (**c**) The top-view sketch of leaf occurrence probability.

To simplify the problem, we make three assumptions: (1) each leaf is a long and narrow rectangle with length *l* and width *W*; (2) leaf azimuth distribution is uniform; and (3) leaf vertical distribution is uniform. Thus, the probability of leaf appearance is inversely proportional to the distance from the central stem, while being proportional to the width of leaf.

The probability of leaves occurrence at one specific distance from the main stem is a classical probability problem (Figure 1c). The probability is inversely proportional to the radius of the circumference at that specific distance from the main stem, and is proportional to the width of leave. The probability can be given by Equation (1):

$$Po(\rho) = nW/(2\pi\rho u) \tag{1}$$

where *Po* is the probability of leaves occurrence, *n* is the number of leaves in this layer, and $\mu$ is the cosine of the leaf inclined zenith angle.

If the thickness of the canopy is *b*, and the number of leaves inserted on a single stem is *N*, then the LAVD at point $(\rho, \varphi, z)$ at individual plant scale can be calculated as follows:

$$Di(\rho, \varphi, z) = NW/(2\pi\rho ub) \tag{2}$$

where *Di* is the LAVD at individual scale, $\rho$ varies in the range from 0 to *l*, *z* varies in the range of [*h*, *H*], and *Di* is 0 when *z* is out of this range.

Because of assumption 1, each leaf is a long and narrow rectangle with length *l* and width *W*, and the area of one leaf is *lW*. Given that the number of leaves inserted on a single stem is *N*, the total leaf area of one individual is given by Equation (3):

$$\int_0^H \int_0^a \int_0^{2\pi} Di(\rho, \phi, z) d\phi d\rho dz = lWN \tag{3}$$

In practice, the special case need to be considered when the radius $\rho$ is close to 0 or when the *Di* is greater than 1 the value of *Di* is set to be 1 artificially.

Given the plant density *M*, we can calculate leaf area index (*LAI*) by Equation (4):

$$LAI = lWNM \tag{4}$$

In practice, the intermediate variables *N, W,* and *M* need not be configured, because they can be substituted by *LAI, L,* and *l*.

Once the LAVD at the individual scale is formulated, the LAVD value at row scale can be obtained by integrating the Di values of several individuals whose leaves can reach the point (Figure 2). Crops are usually planted in line randomly. The LAVD parallel to the row direction can be assumed to be homogeneous. Meanwhile, the LAVD across the row direction is highly heterogeneous. Thus, the three-dimensional question can be simplified to a two-dimensional question. LAVD variation only needs to be considered in the cross-row plane. The LAVD of point $(x,y,z)$ is as follows:

$$Dr(x, y, z) = \int_{y1}^{y2} Di\left( \sqrt{x^2 + y^2}, \phi, z \right) dy \tag{5}$$

where *Dr* is the LAVD at row scale, and the integral range $[y1,y2]$ is $\left[ -\sqrt{l^2 - x^2}, \sqrt{l^2 - x^2} \right]$.

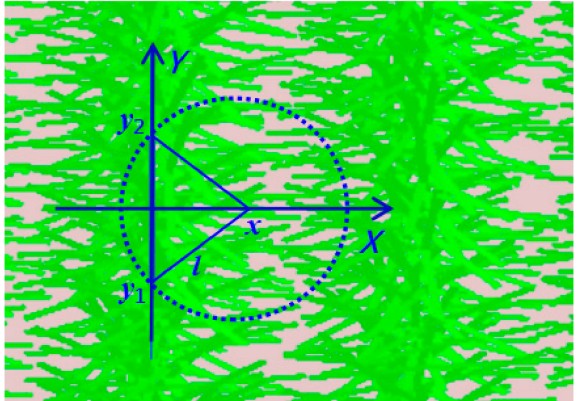

**Figure 2.** Top view of the crop canopy and the integral range for the leaf area volume density (LAVD) at the row scale.

The integral range decreases with the increase of the distance from the central line of the row, while the LAVD at individual scale decreases with the increase of the radius. Therefore, the heterogeneity of LAVD at row scale is the superposition of the above two trends.

Considering periodicity of the row structure, the LAVD at scene scale can be obtained by summation of the LAVD at row scale of neighboring rows.

$$D(x, y, z) = Dr(x, y, z) + Dr(L - x, y, z) \tag{6}$$

where *L* is defined in Figure 1a as the distance between two rows' central line.

### 2.2. Modeling Gap Probability

The gap probability, or frequency of gap, can be defined as the probability that a photon will pass through the canopy unintercepted [38]. It plays an important role in the calculation of radiation transfer in vegetation canopy. Because the transmittance of the leaf is 0 in the thermal infrared (TIR) domain, the transmittance of the whole canopy is approximate to the gap probability.

The gap probability is a function of view direction, LAVD spatial distribution, and leaf inclined angle distribution (LIDF). For homogeneous canopy, the gap probability can be given by the Nilson model [39].

$$P(\theta) = e^{-k(\theta)LAI} \tag{7}$$

$$k(\theta) = \lambda(\theta)G(\theta)/\mu \tag{8}$$

where $P(\theta)$ is the gap probability at a view angle of $\theta$; $\lambda(\theta)$ is the clumping parameter, reflecting the deviation of leaf distribution from random; and $G(\theta)$ is the projection of a unit leaf area onto the plane normal to the view direction. It is the function of LIDF and view direction.

Li et al. (1988) [38] extended the model to be a statistical gap probability model for discontinuous canopy.

$$P_{gap}(\theta) = \frac{1}{A} \iint_A P(x,y,\theta)dxdy \tag{9}$$

where $A$ is the area of the scene and $P(x,y,\theta)$ is the gap probability of point $(x,y)$. When the area of point $(x,y)$ is small enough, the leaves can be seemed as homogeneous, and the clumping parameter $\lambda(\theta)$ is 1. Our new model emphasizes the LAVD spatial variation in row, while row direction apparently influences the gap probability, so the azimuth angle needs be considered. Leaf is nearly randomly distributed in the differential scope. The model is modified to be in the following form:

$$P_{gap}(\theta,\phi) = \frac{1}{A} \iint_A P(x,y,\theta,\phi)dxdy \tag{10}$$

$$P(x,y,\theta,\phi) = \exp(-\int_0^H G(p)D(p)dz) \tag{11}$$

$$\begin{aligned} p &= [x_z, y_z, z] \\ &= [x + z\tan(\theta)\sin(\phi), y + z\tan(\theta)\cos(\phi), z] \end{aligned} \tag{12}$$

The heterogeneous row canopy can be regarded as an assemblage of many sub-rows across the row direction. LAVD decreases gradually from the center of the row to the edge in the plane across rows, while it does not change in the plane parallel rows. Therefore, the above model can be simplified as follows:

$$P_{gap}(\theta,\phi) = \frac{1}{L} \int_0^L P(x,\theta,\phi)dx \tag{13}$$

where $x$ axis is across the row direction. Considering the periodicity of the row structure, the integration range needs only be $[0, L]$.

## 2.3. Bi-Directional Gap Probability

The hot spot effect is a well-known phenomenon in both the reflectance and emittance band. It is the peak in measured radiance in the retroillumination direction. Bi-directional gap probability is defined as the view probability in both view and illumination direction. It is the key way to model the hot spot effect.

There were two models solving the bi-directional gap probability. Kussk's model was widely used for crop canopy [40] and Jupp's model was used for forest canopy [41]. We select Kussk's model to simulate the hot spot effect of the crop canopy. The model is expressed as follows:

$$P_b(r_i, r_v) = P(r_i)P(r_v)C_{HS}(r_i, r_v) \tag{14}$$

where $P_b$ is the bi-directional gap probability; ri and rv are the illumination and view direction, respectively, and include two items, $r = [\theta, \varphi]$; $\theta$ is the zenith angle; and $\varphi$ is the azimuth angle. $C_{HS}$ is the hotspot factor specified as follows:

$$C_{HS}(r_i, r_v) = \exp(\int_0^H \sqrt{\frac{G(r_i, z)G(r_v, z)}{\mu_i \mu_v}} D(z)Y(d)dz) \tag{15}$$

where $H$ is the canopy height; $\mu_i = \cos(\theta_i)$; $\mu_v = \cos(\theta_v)$; and $Y$ is the cross-correlation function, which was statistically expressed with a negative exponential equation:

$$Y(d) = \exp(-d/l_L) \tag{16}$$

$$d = (H - z)\Delta(r_i, r_v) \tag{17}$$

$$\Delta(r_i, r_v) = \sqrt{\mu_i^{-2} + \mu_v^{-2} - 2\cos(\xi)/(\mu_i\mu_v)} \tag{18}$$

where $l_L$ is the mean linear dimension of leaves; $\mu$ is the cosine value; subscripts $i$ and $v$ denote the illumination and view directions, respectively; $\xi$ is the angle between view direction and illuminate direction; and $\cos(\xi)$ can be achieved as follows:

$$\cos(\xi) = \cos(\theta_i)\cos(\theta_v) + \sin(\theta_i)\sin(\theta_v)\cos(\phi_i - \phi_v) \tag{19}$$

In order to adapt to the heterogeneous situation, we extend the model to be a statistical model.

$$P_b(r_i, r_v) = \frac{1}{L}\int_0^L P_b(x, r_i, r_v)dx \tag{20}$$

$$P_b(x, r_i, r_v) = P(x, r_i)P(x, r_v)C_{HS}(x, r_i, r_v) \tag{21}$$

$$C_{HS}(x, r_i, r_v) = \exp\left(\int_0^H \sqrt{\frac{G(p_i, r_i)D(p_i)G(p_v, r_v)D(p_v)}{\mu_i\mu_v}}\right)Y(d)dz \tag{22}$$

where $p$ is the position defined in Equation (12).

### 2.4. Ground-Leaving Directional Radiance (DR)

The thermal infrared (TIR) radiance observed by the ground-based instrument is composed of an emittance item and reflectance item. For the homogeneous case, the spectral radiance sensed by a ground-based detector can be expressed as follows:

$$R_j(\lambda, x, y) = \varepsilon(\lambda, x, y)B(T_s(x, y), \lambda) + (1 - \varepsilon(\lambda, x, y))R_{atm\downarrow}(\lambda) \tag{23}$$

where $R_j(\lambda,x,y)$ is the spectral radiance observed by the spectrometer at the wavelength of $\lambda$ at the position of $(x,y)$; $\varepsilon(\lambda,x,y)$ is the emissivity of the surface $(x,y)$; $T_s(\lambda,x,y)$ is the temperature of the surface $(x,y)$; $B(T,\lambda)$ is the spectral radiance emitted from a blackbody with temperature of $T$ at wavelength of $\lambda$, which can be determined by Planck's law Equation (24); and $R_{atm\downarrow}(\lambda)$ is the downward spectral radiance from the total atmosphere.

$$B(T, \lambda) = \frac{C_1}{\lambda^5\left[\exp(\frac{C_2}{\lambda T}) - 1\right]} \tag{24}$$

where $C_1$ and $C_2$ are two constants for the Planck function.

The heterogeneous surface can be taken as the assemble of homogeneous sub-surfaces. Scattering in the TIR domain is negligible. The radiance emitted from the heterogeneous surface can be calculated by integrating the radiance from each element within the field of view (FOV).

$$R_a(\lambda) = \frac{1}{A}\iint_A R_j(\lambda, x, y)dxdy \tag{25}$$

where $R_a(\lambda)$ is the radiance from the FOV and A is the area of the FOV.

In this paper, the FOV is divided into three components, which are sunlit soil, shaded soil, and foliage. The radiance observed by radiometer is the sum of radiance from each component weighted by its view fraction.

$$R_a(r_v, r_i, \lambda) = \sum_{j=1}^{3} a_j(r_v, r_i) R_j(\lambda) \tag{26}$$

where $R_a(r_v, \lambda)$ is the radiance from the FOV at direction $r_v$ with zenith $\theta_v$ and azimuth $\varphi_v$, $a_j(r_v, r_i)$ is the view fraction of the $j$th component with the view direction $r_v$ and solar direction $r_i$, and $R_j(\lambda)$ is the radiance from the $j$th component.

In the new model, the main cause of directional variation of radiance comes from the view fraction. The view fractions of foliage, sunlit soil, and shaded soil noted are as $a_l(r_v, r_i)$, $a_{sun}(r_v, r_i)$, and $a_{shd}(r_v, r_i)$, respectively. They are determined as follows:

$$a_l(r_v, r_i) = 1 - P_{gap}(r_v) \tag{27}$$

$$a_{sun}(r_i, r_v) = P_b(r_i, r_v) \tag{28}$$

$$a_{shd}(r_i, r_v) = P_{gap}(r_v) - P_b(r_i, r_v) \tag{29}$$

In order to keep the unit of simulation results consistent with other models, directional radiance needs be converted to DBT using the Planck inverse function.

## 3. Results

### 3.1. Evaluation of Forward Simulation

To evaluate the new model's forward simulation performance, the thermal radiosity-graphics combined model (TRGM [32]) is selected to generate forward simulation data as the benchmark. A 3D near-real canopy scene is firstly required to be generated prior to TRGM operation. The scene is composed of individual plants whose leaves are abstracted as narrow and long rectangles stretching from the central stem. Scenes of three growth stages are generated for comparison (Figure 3). Row structure configurations of these scenes are listed in Table 1. In the life cycle of crops, the geometric structure gradually changes from row characteristics to homogeneous characteristics. Stage 1 represents the typical row structure. Stage 2 represents the intermediate period when the leaves of the neighboring rows begin to touch each other. Stage 3 represents the typical homogeneous scene.

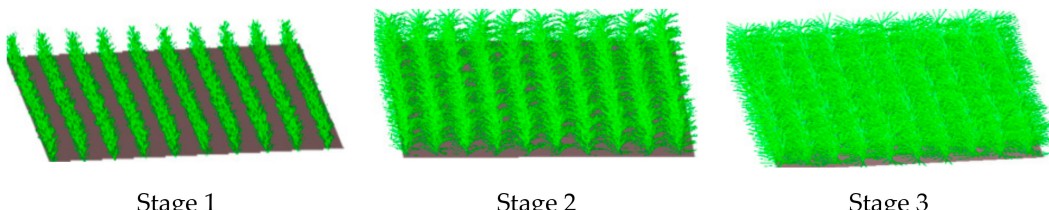

Stage 1　　　　　　　　　　　　Stage 2　　　　　　　　　　　　Stage 3

**Figure 3.** Scenes of three growth stages.

**Table 1.** Row structure configuration of simulation scene for the three growth stages.

|  | $L$ (m) | $H$ (m) | $h$ (m) | $a$ (m) | $\mu$ | $LAI$ |
|---|---|---|---|---|---|---|
| Stage 1 | 0.8 | 0.30 | 0.05 | 0.30 | 0.866 | 0.5 |
| Stage 2 | 0.8 | 1.0 | 0.1 | 0.80 | 0.866 | 1.6 |
| Stage 3 | 0.8 | 1.8 | 0.1 | 1.52 | 0.866 | 2.9 |

In order to highlight the impact of intra-row heterogeneity, the other input parameters are set to be same for the three stages (Table 2). A DBT model proposed by Yu et al. (2004) [1] was used as comparison. Yu's model took into account the gap probability and adopted the formula of hot spot

effect, simulating radiance transfer within the canopy with good simulation accuracy. Yu's model used the clumping index to correct the deviation caused by leaf heterogeneity in rows. The clumping index was first proposed in the V-NIR band [39], and then introduced into the thermal infrared band. Bian et al. (2018) [42] developed a universal DBT model suitable for forest and crop using the clumping index. Chen et al. (2003) [43] found that the normalized difference between hotspot and darkspot (NDHD) is linearly related to the clumping index. This method was adopted to generate the global clumping index [44]. In order to compare with the new model, we set the clumping index to 1, which assumed that the leaves are completely homogeneous. In this way, we simulated the angular distribution of nine groups of DRs for three growth stages using three models (Figure 4).

**Table 2.** Input parameters of simulation scene other than row structure.

| $\theta_i$ | $\varphi_i$ | $T_l$ (K) | $T_{sun}$ (K) | $T_{shd}$ (K) | $\varepsilon_l$ | $\varepsilon_s$ |
|---|---|---|---|---|---|---|
| 20.0 | 140.0 | 300.15 | 318.15 | 306.15 | 0.98 | 0.93 |

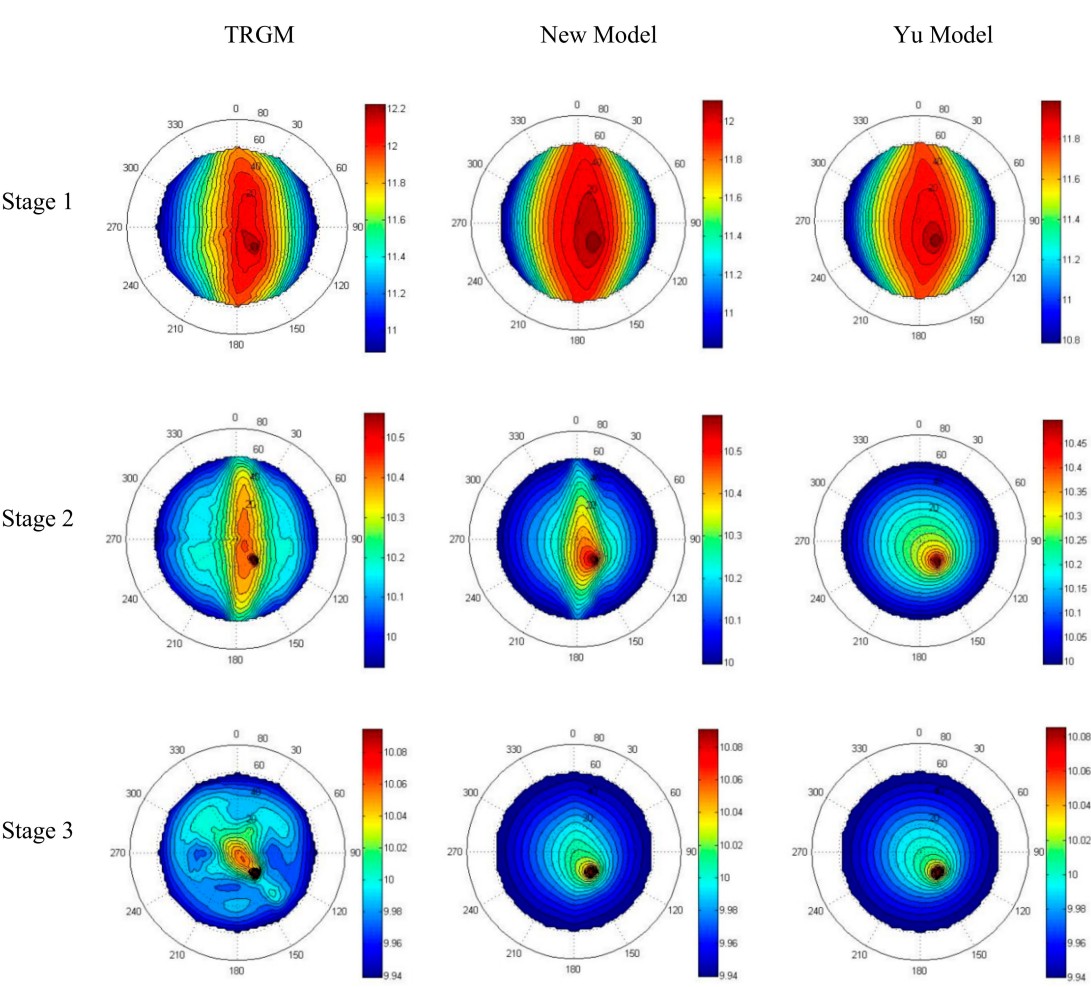

**Figure 4.** Polar plots of simulated directional radiance (DR), units: $w/m^2/sr/\mu m$. TRGM, thermal radiosity-graphics combined model.

At the first growth stage, the scene is made up of alternating plant row and bare soil. More hot soil would be seen when the viewed direction is parallel to the row, resulting in the warm belt occurring in the polar map of the DBT. The simulation results of the new model were very close to those of the Yu model and TRGM model. At the second growth stage, the simulation results of the new model were similar to those of the TRGM model, and had an obvious row effect. The results of the Yu model had no row effect, which were quite different from those of the other two models. At the third growth

stage, the soil was totally shadowed by leaves and the row effect disappeared in the simulation results of the three models. The simulation results were quite close.

In order to further discuss the difference between the new model and Yu model, we plotted the profiles of LAVD crossing row. The LAVD of the new model increases gradually from the edge of row to the center. The LAVD of Yu model increases sharply to a value at the edge of row, and then remains constant inside row. When the width of row equals the distance between two rows at growth stage 2, LAVD of Yu model is uniform over the whole field, while LAVD of the new model fluctuated greatly from 0 to 5 Figure 5(a2). At growth stage 3, LAVD of the new model is nearly uniform with a little peak in the center Figure 5(a3).

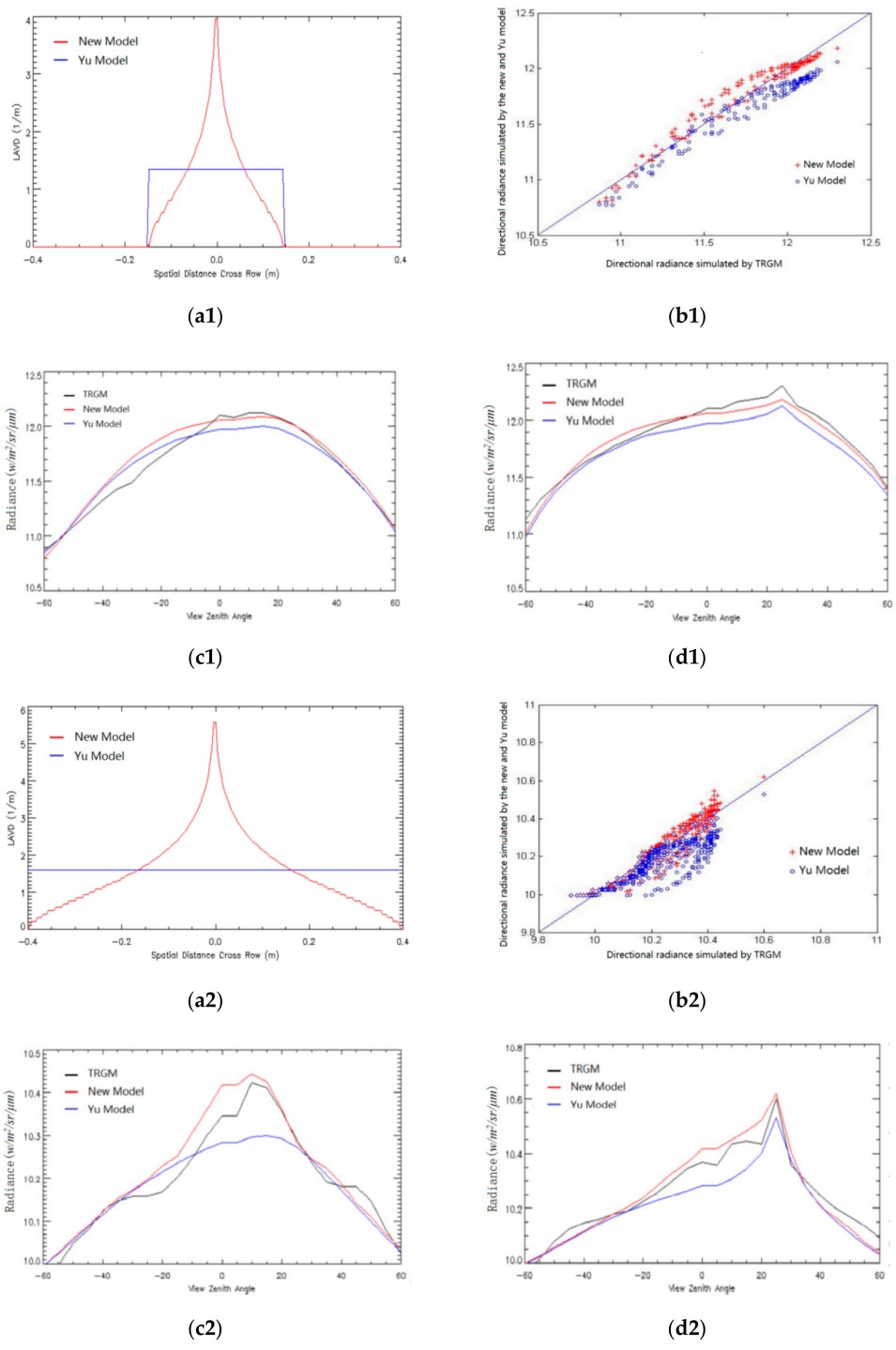

**Figure 5.** *Cont.*

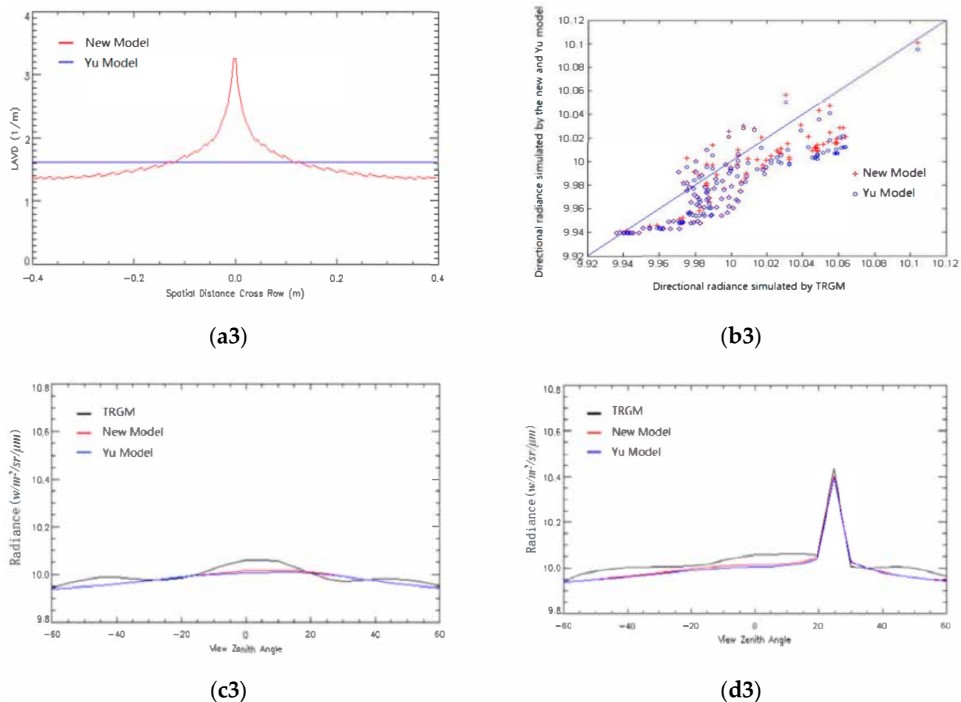

**Figure 5.** (**a1**), (**a2**), (**a3**) The LAVD profiles crossing row simulated by the new model (red line) and Yu model (blue line) for growth stage 1, 2, and 3 separately. (**b1**), (**b2**), (**b3**) The scatter plots for the DRs simulated by new model (red dots) and those by Yu model (blue dots) against those by TRGM for growth stage 1, 2, and 3 separately. (**c1**), (**c2**), (**c3**) The DBTs simulated by TRGM (black line), the new model (red line), and Yu model (blue line) for growth stage 1, 2, and 3 separately in the crossing row plane. (**d1**), (**d2**), (**d3**) The DBTs simulated by TRGM (black line), the new model (red line), and Yu model (blue line) for growth stage 1, 2, and 3 separately in the solar principle plane (SPP).

In the crossing row plane (CRP) and the solar principle plane (SPP), the simulation results of the three models are in good agreement at stage 1 and 3 Figure 5(c1,d1,c3,d3). The simulation results of the new model are a little higher than those of Yu model. Because the heterogeneity is taken into account, canopy gap probability increases, resulting in more radiation from hotter soils being observed. At growth stage 2, the simulation results of Yu model are apparently lower than those of the new model and TRGM model near Nadir Figure 5(c2,d2).

*3.2. Evaluation of CT Retrieval Ability*

Our concern is not only the forward simulation of DBT, but also the retrieval of CT, which is significant for estimating evapotranspiration using a two-source energy balance model [35]. To evaluate the model's ability of CT inversion, three groups of simulation have been done using TRGM model generated synthetic data for the three growth stages, as mentioned above. The new model was used to retrieve CT from the simulated multi-angle thermal radiance. As a comparison, Yu model was also used to retrieve CT.

Thermal radiance varies with view angle because of the differences in component emissivity, component temperature, and proportion of the components Equation (30). The principle of CT inversion is to solve linear equations of CT, component directional emissivity, and directional radiance [45].

$$R = WB_{ct} \tag{30}$$

$$
R = \begin{bmatrix} R(\theta_1) \\ R(\theta_2) \\ \vdots \\ R(\theta_m) \end{bmatrix} \quad B_{tc} = \begin{bmatrix} B(T_1) \\ B(T_2) \\ \vdots \\ B(T_3) \end{bmatrix} \tag{31}
$$

$$
W = \begin{bmatrix} \varepsilon_1(\theta_1), \ \varepsilon_2(\theta_1), \ \cdots, \ \varepsilon_n(\theta_1) \\ \varepsilon_1(\theta_2), \ \varepsilon_2(\theta_2), \ \cdots, \ \varepsilon_n(\theta_2) \\ \vdots \\ \varepsilon_1(\theta_m), \ \varepsilon_2(\theta_m), \ \cdots, \ \varepsilon_n(\theta_m) \end{bmatrix} \tag{32}
$$

where $R$ is the measured directional radiance vector after atmospheric correction, $B_{tc}$ is the blackbody thermal radiance vector of CTs, and $W$ is the matrix of component effective emissivity [46]. In this paper, $R$ is simulated by TRGM at three view angles in the SPP, including Nadir, hotpot direction, and zenith of 50 degree. $W$ can be obtained by the new model or Yu model. $B_{tc}$ is solved by the least-square method.

The inversion results are shown in Figure 6. The accuracy of leaf temperature inversion is good at each stage. The accuracy of soil temperature inversion varies greatly in different stages. At growth stage 1, the inversion accuracy of the two models is comparable. The inversion accuracy of sunlit soil temperature is higher than that of shaded soil temperature (Figure 6a). At growth stage 2, the new model has an obvious advantage for soil temperature inversion (Figure 6b); the RMSEs are 2.3 K and 1.8 K for sunlit and shaded soil, separately, which is better than Yu model, whose RMSEs are 4.7 K and 4.6 K, correspondingly (Table 3). At stage 3, the accuracy of the two models for soil temperature is very low, with RMSE greater than 5 K (Figure 6c).

Errors of CT inversion come from three sources, including simulation uncertainties of TRGM, errors of DBT model to generate the emissivity matrix, and errors of equation solutions. The RMSE of TRGM simulation was examined to be around 0.5 K [32]. The inversion accuracy of the new model is apparently better than that of Yu model at stage 2 because the new model generates a better emissivity matrix than the Yu model (Figure 6b). At stage 3, when LAI is 2.9, too little information of soil temperature can be measured, resulting in instability of the equation solution and bad inversion accuracy of the soil temperature (Figure 6c).

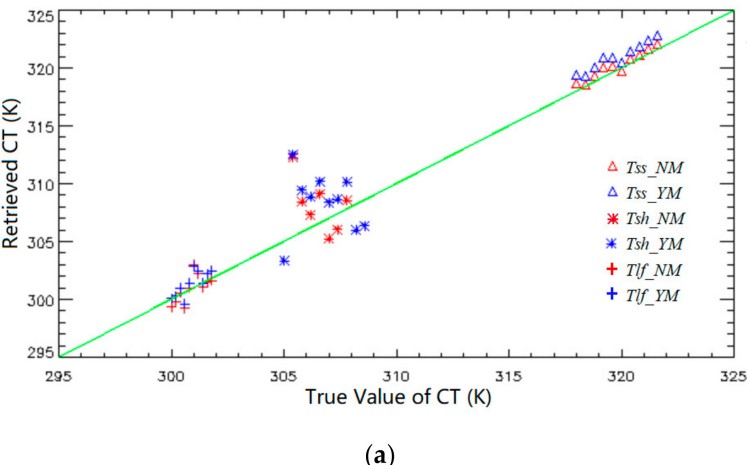

(**a**)

**Figure 6.** *Cont.*

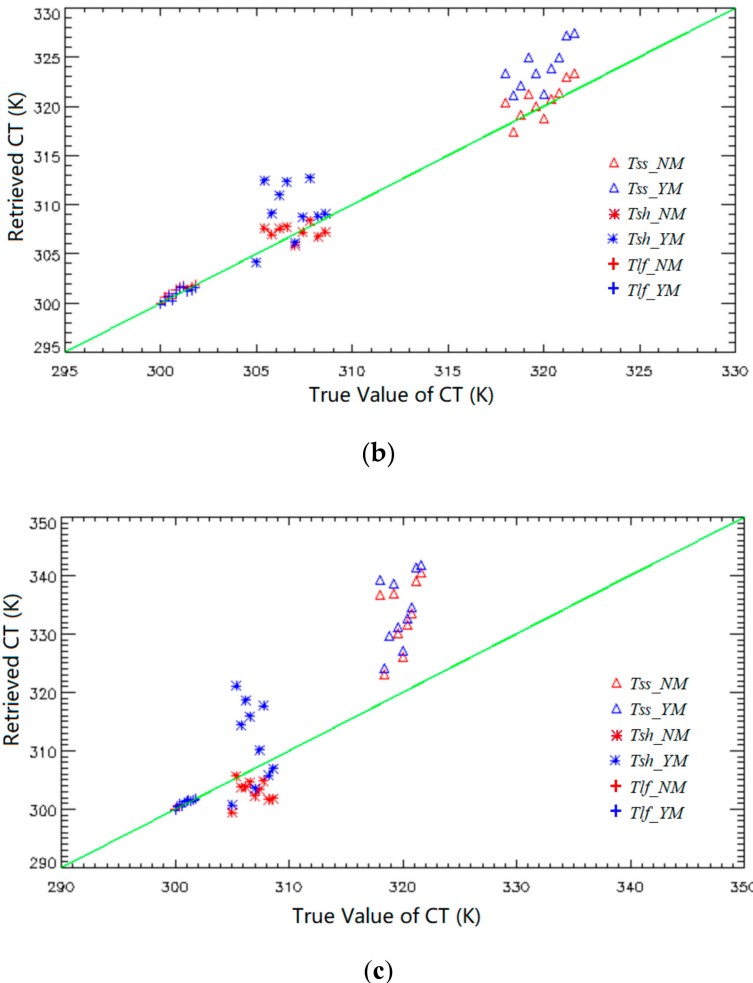

**Figure 6.** The results of component temperature (CT) inversion. (**a**) Growth stage 1, (**b**) growth stage 2, and (**c**) growth stage 3. *Tss_NM* and *Tss_YM* represents the sunlit soil temperature retrieved by the new model and the Yu model separately. *Tsh_NM* and *Tsh_YM* represents the shaded soil temperature retrieved by the new model and the Yu model separately. *Tlf_NM* and *Tlf_YM* represents the leaf temperature retrieved by the new model and the Yu model separately.

**Table 3.** RMSE of component temperature (CT) inversion at the three growth stages.

| RMSE of CT (K) | Stage 1 | Stage 2 | Stage 3 |
|---|---|---|---|
| $\Delta Tss\_NM$ | 0.5 | 2.3 | 11.3 |
| $\Delta Tss\_YM$ | 0.8 | 4.7 | 12.5 |
| $\Delta Tsh\_NM$ | 3.3 | 1.8 | 4.6 |
| $\Delta Tsh\_YM$ | 3.4 | 4.6 | 8.7 |
| $\Delta Tlf\_NM$ | 1.3 | 0.8 | 0.5 |
| $\Delta Tlf\_YM$ | 1.2 | 0.7 | 0.6 |

*3.3. Row Effect Spanning Growth Season*

In Sections 3.1 and 3.2, we select three growth stages to discuss the forward simulation and inversion capabilities of the new model. Crop growth is a continuous process. In this section, we will discuss the performance of new model throughout the whole growth season.

In the early stage, row planting crops will show a strong row effect, that is, DBT or BRDF not only varies with view zenith angle, but also varies with the view azimuth angle and is influenced by row orientation. In the later stage, crops can be considered as continuous vegetation, which allows a one-dimensional model to do simulation or inversion [47].

Before discussion, we firstly define a row effect index (REI) as a ratio of the difference between off-nadir DBT and nadir DBT in the along row plane (ARP) and the difference in the crossing row plane (CRP) Equation (33).

$$REI = (DBT(\theta_1, \varphi_1) - DBT(0,0))/DBT(\theta_1, \varphi_2) - DBT(0,0)) \tag{33}$$

where $\theta_1$ is the off-nadir zenith angle. $\varphi_1$ is the azimuth angle of ARP. $\varphi_2$ is the azimuth angle of CRP. In this paper $\theta_1$, $\varphi_1$, and $\varphi_2$ are set as 40, 0, and 90, separately. Row structure parameters are configured in Table 4. Other parameters such as illuminate angle, CT, and component emissivity are listed in Table 2.

**Table 4.** Row structure configuration for row effect index (REI) spanning growth season.

|  | **Min Value** | **Max Value** | **Step Value** |
|---|---|---|---|
| $H$ (m) | 0.2 | 2.0 | 0.2 |
| $h$ (m) | 0.01 | 0.1 | 0.1 |
| $a$ (m) | 0.1 | 1.0 | 0.1 |
| $LAI$ | 0.3 | 3.0 | 0.3 |
| $\mu$ | 0.866 | 0.866 | 0 |
| $L$ (m) | 1.0 | 1.0 | 0 |

We select the ratio of row width $a$ and row distance $L$ as the $X$ axis to plot the REIs. At the beginning, the REIs of the two models are very similar. In the case of neglecting intra-row heterogeneity, REI of the Yu model increased rapidly and reached 1 when $a/L$ equals 0.5. Considering intra-row heterogeneity, REI of the new model slowly close to 1 (Figure 7).

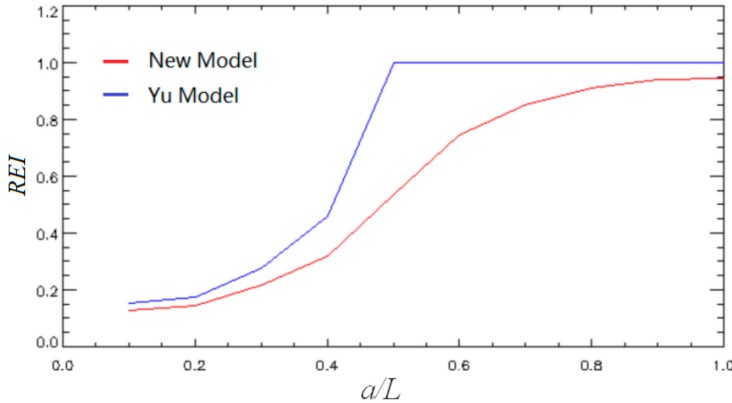

**Figure 7.** Row effect index (*REI*) changes with row structure configurations.

*3.4. Forward Simulation Spanning Growth Season*

The difference between the new model and Yu model can reflect the influence of intra-row heterogeneity. We want to know how the simulation difference between the two models changes with the growth season. We chose three directions, including nadir, hotspot, and 50° in SPP. The input configuration was the same as Tables 2 and 4. The simulation difference at off-nadir 50° remains relative small compared with that of nadir or hotspot. The simulation difference at nadir and hotspot direction changes greatly with $a/L$. When $a/L$ is 0.5, the simulation difference is the greatest (Figure 8).

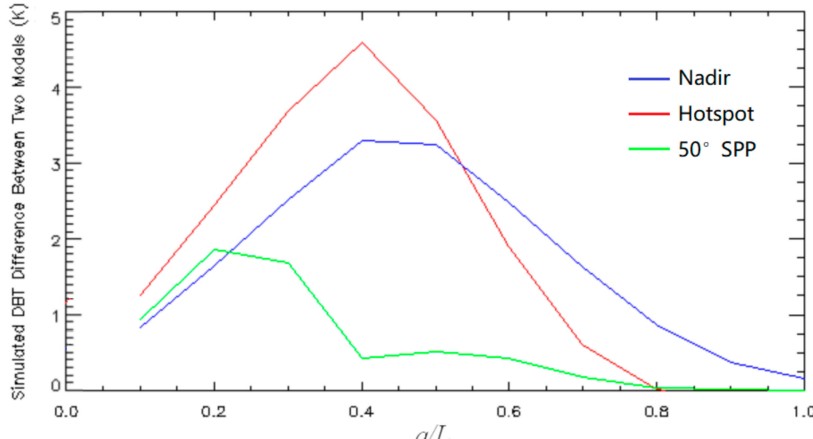

**Figure 8.** Directional brightness temperature (*DBT*) simulation difference between the new model and Yu model changes with row structure configurations.

*3.5. CT Retrieval Spanning Growth Season*

We used the parameters of Tables 2 and 4 as the inputs of TRGM to simulate the directional radiation at three angles in SPP, including nadir, hotspot, and zenith of 50°. We use the method introduced in Section 4 to retrieve the CTs. The absolute difference between retrieved CTs and input CTs was taken as the inversion error. Thus, the trend of CT inversion error varying with structural parameters can be obtained (Figure 9).

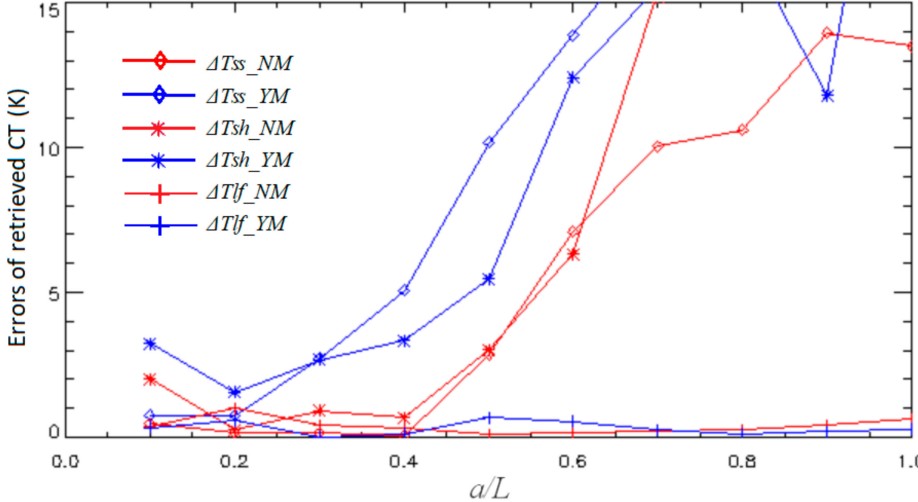

**Figure 9.** Errors of retrieved CT change with row structure configurations. Δ*Tss_NM* and Δ*Tss_YM* are errors of sunlit soil temperature inversion by the new model and by the Yu model, respectively. Δ*Tsh_NM* and Δ*Tsh_YM* are errors of shaded soil temperature inversion by the new model and by the Yu model, separately. Δ*Tlf_NM* and Δ*Tlf_YM* are errors of leaf temperature inversion by the new model and by the Yu model, separately.

As can be seen from Figure 9, the error of leaf temperature inversion is around 1 K around in all cases. At the beginning of the growing season, the inversion errors of CT between the two models are compatible. In the middle stage, the accuracy of soil temperature inversion by the new model is better than that by Yu model. In the later stage, the soil temperatures retrieved by the two models are both unreliable, with errors greater than 10 K owing to the too weak information from soil with the high LAI.

## 4. Discussion

*4.1. Sensitivity Analysis of Input Parameters*

A single-factor sensitivity analysis was conducted to examine the response of the simulated DR to each input parameter. In each analysis, only one input parameter would be changed and other inputs kept unchanged (the same as that of stage 2 in Table 1).

We changed the value of LAI from 0.5 to 3.0 at increments of 0.5, and the simulated DR decreased owing to the increased leaves blocking the radiance from the warm soil surface Figure 10(a1,b1). We changed the width of row (*a*) from 0.2 m to 1.2 m at increments of 0.2 m. When the crop row is narrow, more bare soil is exposed to sunlight. That is why so high simulated DR values were obtained for the low row width in the near-nadir angle. In this case, the "row effect" is strong. As the row width was increased, DR simulated in the near-nadir angle decreasing owing to the warm soil can not be seen directly, and the "row effect" is dimmed Figure 10(a2,b2). Low DR in the near-nadir angle makes the hot spot effect remarkable Figure 10(b2). Increases in the height of the upper edge of the row (*H*) from 0.6 m to 1.35 m at increments of 0.15 m had a decreasing effect on the simulated DR. Low vegetation rows allow more sunlit soil in the gap between rows. With the *H* increasing, sunshine would be blocked by the high vegetation Figure 10(a3,b3). The contrary was observed for the height of the bottom edge of the row (*h*). Increments of 0.06 m from 0.0 m to 0.3 m had an increasing effect on the simulated DR Figure 10(a4,b4). With the *h* increasing, the vegetation rows become thinner and allow more sunshine to penetrate the gap between rows. When the CT of shaded soil increases to be close to that of sunlit soil, the simulated DR increases Figure 10(a5) and the hot spot effect disappears Figure 10(b5). The simulated DR increases with the CT of leaves in all view angles Figure 10(a6,b6) as the leaves account for the greatest view fraction in all view angles.

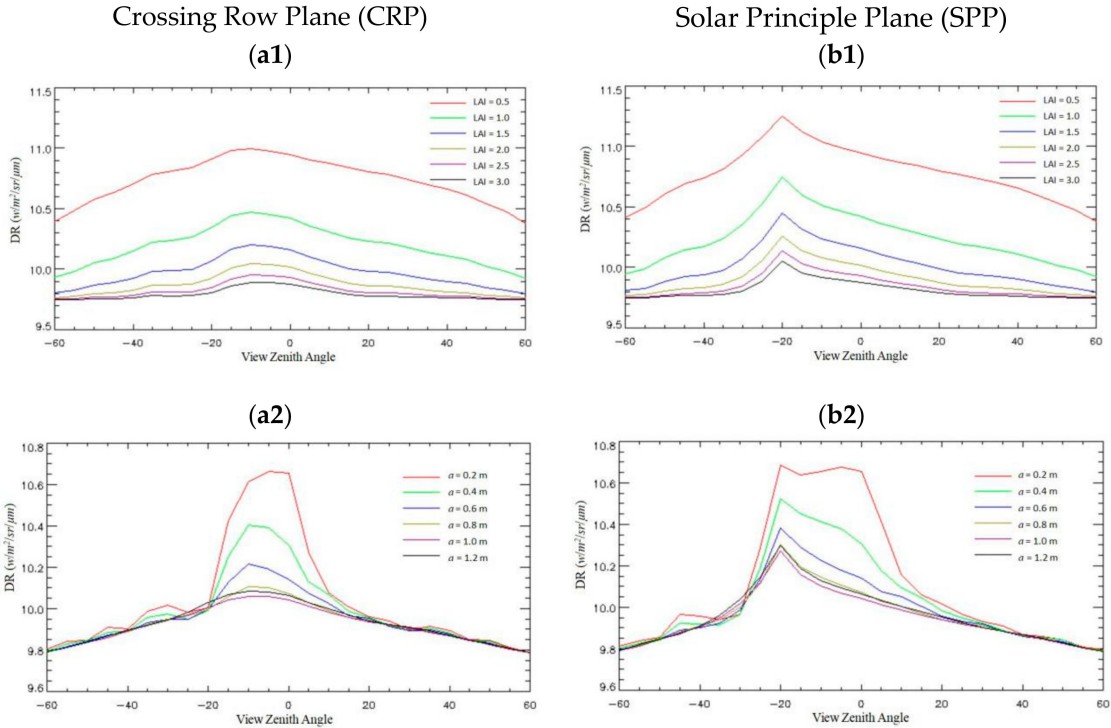

**Figure 10.** *Cont.*

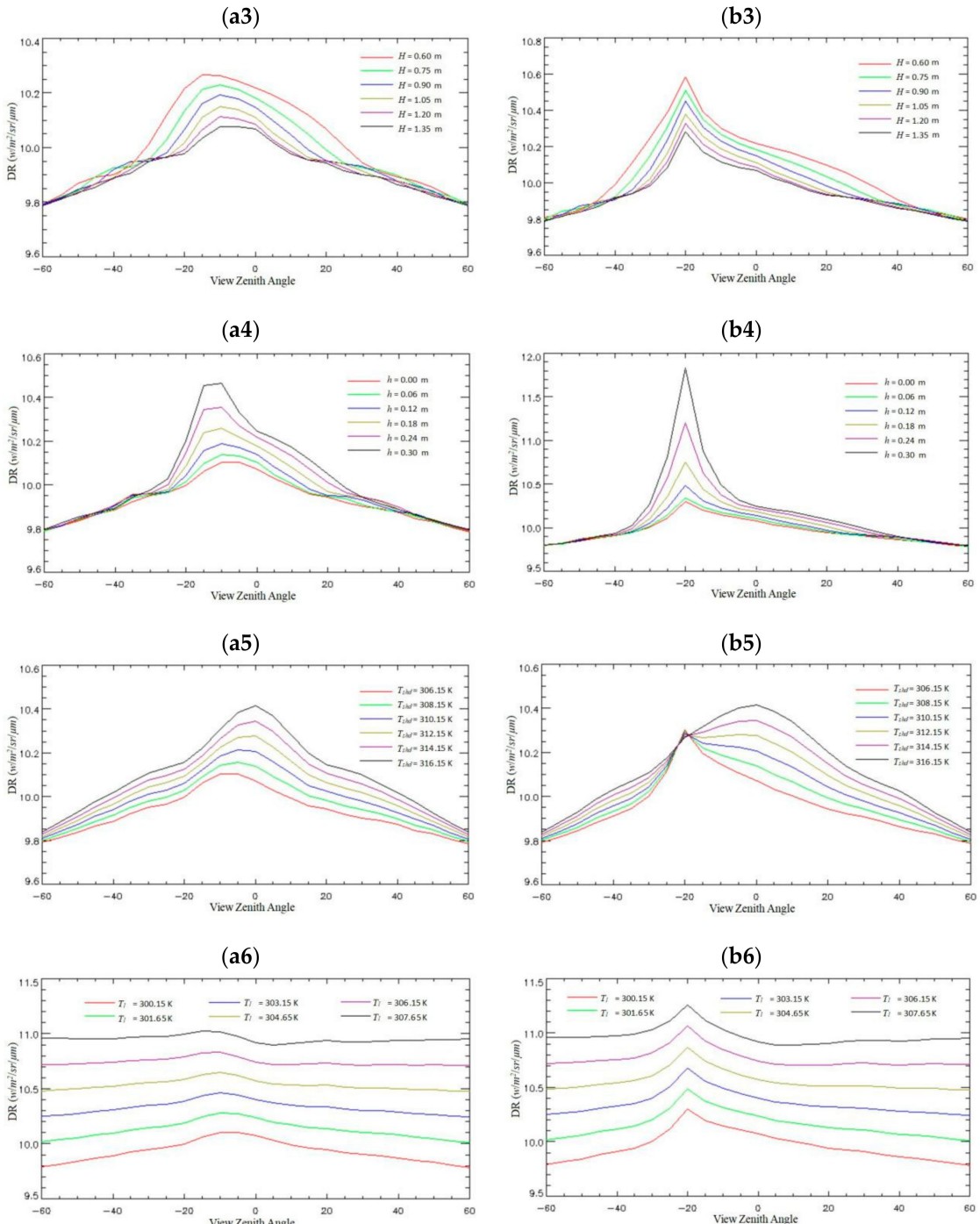

**Figure 10.** Sensitivity analysis of input parameters. (**a1**) and (**b1**) Sensitivity analysis of LAI in crossing row plane (CRP) and solar principle plane (SPP), respectively. (**a2**) and (**b2**) Sensitivity analysis of row width in CRP and SPP, respectively. (**a3**) and (**b3**) Sensitivity analysis of height of upper edge of row in CRP and SPP, respectively. (**a4**) and (**b4**) Sensitivity analysis of height of bottom edge of row in CRP and SPP, respectively. (**a5**) and (**b5**) Sensitivity analysis of CT of shaded soil in CRP and SPP, respectively. (**a6**) and (**b6**) Sensitivity analysis of CT of leaf in CRP and SPP, respectively.

## 4.2. Potential Applications

The directional heterogeneity brings errors in TIR remote sensing applications for monitoring crops. A proper model can minimize this error. The geometric structure of row crops changed greatly during its life cycle. Although there are DBT models for ridge row and continuous vegetation, they still

can not meet the simulation requirements of different growth periods. The capacity of the new model to describe the heterogeneity within the ridge row donates it with the ability to simulate the directional radiance at different growth stages. The new model can be used to analyze and correct the directional variations of thermal radiance leaving from the crop canopy.

TIR remote sensing is widely used to characterize crop water status of crop using indices of CWSI, WDI, and so on. Although it is widely accepted that the directional heterogeneity of thermal radiance affects the accuracy of these indices, there is still no way to correct their angular variations. The lack of proper model may be one of the reasons. The new model supplies a tool to mimic the directional radiance of crop canopy with good accuracy. It can be used to correct the angular variation of CWSI, WDI, and so on.

The estimation of evapotranspiration is of great value for farmland water resources management. The CT provides more accurate energy state information than radiative temperature. The new model can improve the estimation accuracy of CT. This CT retrieval method can be taken into practice to estimate evapotranspiration. Furthermore, the improved CT estimation can be further applied for calculating thermal inertia, which is subsequently used to derive soil moisture content over the non-vegetated soil surface.

### 4.3. The Model's Limitation

The model assumes that there is a gradually changing LAVD in the cross-row direction of the row, which is close to the actual situation compared with the homogeneous assumption. Nevertheless, crop leaves are heterogeneous in both the vertical and azimuth distribution. The shape of leaf is simplified as a rectangle, which is obviously not true. Therefore, the future research along this line is how to optimize the model to make it approach the real situation.

### 5. Conclusions

In this paper, a DBT model considering the intra-row heterogeneity is proposed. This new model formulated the trend that the probability of leaf appearance decreases gradually from center to edge, and simulates LAVD spatial distribution at three scales, including the individual scale, row scale, and scene scale. The equations for directional gap probability and bi-directional gap probability were modified to adapt the heterogeneity of row structure. Afterwards, a straightforward radiative transfer model was build upon the gap probabilities.

To examine the forward simulation and inversion capacity of the new model, TRGM is used to generate synthetic benchmark data. The DBT simulated by the new model still shows strong azimuth heterogeneity when leaves begin to overlap between neighboring rows, which is consistent with the simulation results of TRGM model. In contrast, the Yu model cannot simulate the row effect at this stage owing to its lack of considering the intra-row heterogeneity. In early stage and later stage, the new model has similar accuracy with the Yu model. In the middle stage, the new model performs much better than the Yu model not only in forward simulation, but also in CT inversion. As crop growth is a continuous process, it is difficult to distinguish which stage is suitable for a specific model. We recommend the new model for DBT simulation and CT inversion of row crops. Furthermore, the LAVD modeling method proposed in this paper is not only suitable for thermal infrared domain, but also has potential application for modeling bidirectional reflectance distribution function (BRDF) in the V-NIR domain.

**Author Contributions:** Y.D. proposed the research methodology, designed and performed the evaluation, and wrote the manuscript. B.C. contributed to the TRGM model simulation. H.L. and Z.B. contributed to the CT retrieval. B.Q. contributed to the data processing. Q.X., Q.L., Y.Z., and Z.S. had great contributions on the manuscript review and editing. All authors have read and agreed to the published version of the manuscript.

**Funding:** This research was funded in part by the National Natural Science of Foundation of China, grant number 41930111, 41571359, and 41871258; in part by the Youth Innovation Promotion Association CAS, grant number 2020127; and in part by the "Future Star" Talent Plan of the Aerospace Information Research Institute of Chinese Academy of Sciences, grant number Y920570Z1F.

**Conflicts of Interest:** The authors declare no conflict of interest.

**Abbreviations**

| | |
|---|---|
| TIR | Thermal Infrared |
| LAI | Leaf Area Index |
| TRGM | Thermal Radiosity-Graphics combined Model |
| LST | Land Surface Temperature |
| CWSI | Crop Water Stress Index |
| ET | Evapotranspiration |
| RT | Radiative Transfer |
| NDHD | The Normalized Difference between Hotspot and Darkspot |
| CRP | Crossing Row Plane |
| V-NIR | Visual and Near Infrared |
| LAVD | Leaf Area Volume Density |
| DBT | Directional Brightness Temperature |
| CT | Component Temperature |
| WDI | Water Deficit Index |
| TSEB | Two Source Energy Balance Model |
| GO | Geometric Optics |
| SPP | Solar Principle Plane |
| FOV | Field of View |

**Nomenclature**

| | |
|---|---|
| $L$ | The distance between rows |
| $b$ | The thickness of row |
| $h$ | The heights of upper and bottom edge of row |
| $W$ | The width of leaf |
| $\lambda(\theta)$ | The clumping parameter |
| $P_b$ | The bi-directional gap probability |
| $\theta$ | The zenith angle |
| $a$ | The width of row |
| $H$ | The heights of upper and bottom edge of row |
| $l$ | The length of leaf |
| $\mu$ | The cosine of zenith angle |
| $P(\theta)$ | The gap probability at view angle of $\theta$ |
| $C_{HS}$ | The hotspot factor |
| $\varphi$ | The azimuth angle |

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
