# Peer review of "Modeling Directional Brightness Temperature (DBT) over Crop Canopy with Effects of Intra-Row Heterogeneity"

_remotesensing, doi:10.3390/rs12172667_

Round 1

Reviewer 1 Report

The authors present a model for simulating directional brightness temperature, taking heterogeneities within rows into account. Evaluation of the model was done by comparison with existing models, in forwarding simulation, and by testing the performance at different growth stages.

The introduction is well written. An overview of existing publications and the current state of science is given. The section gives enough background information for understanding the authors’ motivation and the relevance of their approach. It becomes clear that their new model has the potential for improving knowledge within TIR remote sensing.

Following the introduction, the authors provide the following structure: (2) Description of the model – (3) Evaluation of forwarding simulation – (4) Evaluation of CT retrieval ability – (5) Discussion – (6) Conclusion. To me, this structure is quite strange, and also it does not meet the requirements given in the instructions for authors.  In section (2) – which should be “Materials and Methods” – the model is introduced. Sections (3), (4) and (5) are one continuum and should be considered as “Results”. Most importantly, section (5) is an extended presentation of results and does not fulfill the requirements of a discussion (i.e. discuss the results and how they fit into existing literature, discuss limitations of the new model, discuss future directions of research, etc.). The discussion is clearly the weakest part of the manuscript and needs proper revision. The discussion section does only contains 1 (!) citation. After giving a nice introduction, it should be possible for the authors to provide a proper and critical view of their results. The last part (6) is well written, still in a conclusion, it should not be necessary to refer to multiple figures. The conclusion should be self-explanatory after the main points were elaborated in the discussion.

Specific comments

Overall, the language of the manuscript is fine, though there are some typographical errors and some sentences needing revision (e.g. lines 80/81, line 92, line 144).  

Line 22: TRGM is introduced as an unexplained abbreviation.

Section 2, Fig. 1. Did the authors have any specific crop species in mind? The drawing looks like maize. Also, may future approaches use species-specific variations of the model?

Section 3, Tab 1. and Fig. 3 Do the numbers in the first column of Tab.1 correspond to the labeling (a, b, c) of the growth stages in Fig. 3? What was the rationale for picking exactly these growth stages? Do they represent a developmental stage of special interest?

Section 3, Fig. 4. I like this visualization, however, the labeling of the subplots may be improved. How about titles for each column (TRGM – New model – Yu model) and row (stage 1- stage 2 – stage 3)?

Section 3, Fig. 5 Some of the plots are missing y-axis labeling. In some aspects, the figure is tough to digest the figure, as scatter plots were combined with line graphs. You may consider splitting up this figure into two separate graphs.

Author Response

Dear Reviewer,

    Thank you very much for the time you spent on our manuscript and for giving us the comments to improve the work. We took the comments and suggestions seriously and addressed each of them in detail. We hope these revisions resolve the problems and uncertainties noted by you.  A point by point response is included in the ms-word file below. Thank you again. 

Reviewer 2 Report

I recommend extending the chapter 5 Discussion of other authors. Overall the manuscript is at a good level and can be published.

Author Response

Dear Reviewer,

    Thank you very much for the time you spent on our manuscript and for giving us the comments to improve the work. We took the comments and suggestions seriously and addressed each of them in detail. The type front you noted have been corrected. We hope these revisions resolve the problems and uncertainties noted by you. Thank you again.  

Reviewer 3 Report

Agricultural crops are important targets for remote sensing instruments. Agricultural crops are usually planted in rows, which means that, for a considerable time during the early part of the growing season, the stripes of bare soil between rows form an important contribution to the sensor. The authors proposed a new model by considering intra-row heterogeneity to simulate the directional brightness temperature over row crops. The paper is clearly written and is appropriate for publication in RS.

I recommend the acceptance of the manuscript after minor revisions. Specific comments: lines 106-107, can the authors elaborate on why the three assumptions support this conclusion? line 109, ρ is not explained. Can the authors explain how Eq. 1 is derived? line 118, in Eq. 3, z changes from h (not 0) to H. line 122, I am confused by the units here: lWN is in a unit of m2, M (plant density) is in the unit of number/m2, right? Then unit of lWNM is m2*number/m2 = number. But we know the unit of LAI is m2/m2. They are different. lines 123-124, can the authors elaborate the transformation between N, W, and M and LAI, L and l? line 153, Eq. 8, note μ is already used in Eqs. 1&2. line 169, along->cross? lines 259-265, the row effect is a feature for row crops. I suggest the authors explain more how it is formed and compare it to that in the visible, near infra-red band, e.g. as cited in ref. 31,
https://doi.org/10.1016/j.rse.2009.09.018. Fig. 4, second row, c3->c2. Fig. 9, some points are out of the range.

Author Response

Dear Reviewer,

Thank you very much for the time you spent on our manuscript and for giving us the comments to improve the work. We took the comments and suggestions seriously and addressed each of them in detail. We hope these revisions resolve the problems and uncertainties noted by you. A point by point response is listed in the ms-word file below. Thank you again.  

Round 2

Reviewer 1 Report

Dear authors,

Thank you very much for submitting a revised version of your manuscript, as well as for addressing my comments in detail. I now recommend the manuscript to be accepted in the present form.

Author Response

Thank you for the time spent on our manuscript.

In the new edition, we added sensitivity analysis in Section 4 Discussion.